# Triggering Rainfall of Large-Scale Landslides in Taiwan: Statistical Analysis of Satellite Imagery for Early Warning Systems

Tsai-Tsung Tsai *, Yuan-Jung Tsai, Chjeng-Lun Shieh and John Hsiao-Chung Wang

Department of Hydraulic and Ocean Engineering, National Cheng Kung University, Tainan 701, Taiwan
* Correspondence: victor@dprc.ncku.edu.tw; Tel.: +886-6-2091061 (ext. 10)

**Abstract:** Typhoon Morakot had a serious impact on Taiwan, especially the uncommon type of landslide called large-scale landslide (LSL), not many in number but serious in effect, the origin of which the study induced. To establish a specific relationship between LSL and triggering rainfall for future applications of LSL early warning predictions, relevant cases from satellite imagery, along with field investigation data, major event reports, and seismic data from 2004 to 2016, were collected. All collected cases are distributed around the mountainous area in Taiwan, and a total of 107 cases which were mainly distributed in the southern part of the mountainous area were finally selected, including 28 occurrence-time-known cases and 79 occurrence-time-unknown cases. In addition, 149 potential areas identified by the Soil and Water Conservation Bureau (SWCB) were used for improving bounding estimates. Based on the concept of safety factor, two dimensionless quantities, rainfall/landslide depth ($R/D$) and friction angle/slope ($\phi/\theta$), were analyzed by linear regression. In addition, $D$ was assumed to be nonlinearly dependent on $R$, $\theta$, and $\phi$, and the parameter uncertainties were evaluated by the resampling with bootstrap method. Based on the currently obtained data, there were 8% Type-I errors in the results of the linear regression analysis, and 1% Type-II errors in the results of the nonlinear regression analysis. Through the comparison of statistical indicators, the results of nonlinear regression analysis have a better correlation trend. Based on the needs of early warning operations, more conservative indicators can reduce the risks faced by management operations. Therefore, according to the results of this study, the lower boundary values from nonlinear analysis could be used as the LSL early warning management settings. Incorporated with real-time rainfall forecasts, the variation of statistical indicators will provide the trend information dynamically, and will help to increase the response time for relevant evacuation operations, that will be welcome for the further extended applications to guide the evacuation operations of early warning systems.

**Keywords:** landslide; large-scale landslide; triggering rainfall; early warning system; linear regression; nonlinear regression; uncertainty

## 1. Introduction

Typhoon Morakot brought Taiwan a historically record-breaking rainfall in August 2009. Half of Taiwan Island had accumulated rainfalls exceeding 500 mm [1] with some hot areas receiving up to 3000 mm [2]. There were 673 people killed, 26 people missing, and more than TWD 19.5 billion in agricultural losses [3]. Various types of sediment disasters occurred in large numbers during Typhoon Morakot.

Among the sediment disasters, there were 43,519 landslide cases found after this event, but only 259 cases of the uncommon type of landslide called large-scale landslide (LSL), which is not many in number but serious in effect, were found. The most famous case was Xiaolin Village, in Jiaxian District of Kaohsiung City, which was horribly destroyed in this event and 462 people from 180 households were buried in mud and rocks [4]. The tragedy of Xiaolin Village not only shocked all sectors of society, but also made the public realize

that we are not familiar with LSL, and knowledge of the key for prevention and mitigation of LSL incidents is urgently needed. This need prompted the present study.

After Typhoon Morakot, the National Science and Technology Center for Disaster Reduction (NCDR) of Taiwan defined an LSL as a landslide area larger than 10 ha, earth volume larger than 100,000 m$^3$, or a collapse depth deeper than 10 m [5]. Under the situation that epistemic conditions of LSLs are insufficient, in order to find a practical solution for early warning works, this study tries to find the most intuitive indicator, and rainfall and some other factors of LSLs seem to be the best choice. In order to establish the specific relationship between LSLs and triggering rainfall for the future LSL early warning predictions, LSL cases, satellite imagery, rainfall data, seismic data, and other support datasets were collected. In this study, two dimensionless factors, rainfall/landslide depth ($R/D$) and friction angle/slope ($\phi/\theta$), were assumed to have a linear relationship, and all factors, $R$, $D$, $\phi$, $\theta$, were assumed to have a nonlinear relationship, and the both linear and nonlinear regressions were analyzed statistically.

The results of LSL occurrence time and location evaluated from the data of the Broadband Array in Taiwan for Seismology (BATS) were applied in this study for establishing the specific relationship between LSLs and triggering rainfall. Compared with previous research results using only cumulative rainfall of events, the reliability is expected to be considerably improved.

## 2. Literature Review

Current research topics of LSLs include a wide range of subjects, e.g., occurrence mechanisms, monitoring, and early warning. In order to clearly understand past studies, including the key points and effectiveness, a detailed literature review on the above topics was conducted.

### 2.1. Occurrence Mechanism-Related Research

The stability of a slope can be considered by the factor of safety (FS), the ratio of shear strength $\tau_R$ and shear stress $\tau_D$. When the FS is greater than or equal to 1, it can be regarded as a balanced state [6]. The limit equilibrium method is popular in many studies on slope stability analysis [7]. Slope stability analyses can be roughly divided into three common theories based on the corresponding concept of slope type: infinite slope theory [6,8–11], finite slope theory [6,9,12], and the method of slices [6,9–11].

Numerous numerical models have been developed based on the three simulation functionalities of movement status as the following:

1.  Simulate the occurrence of landslide only models, such as SHALSTABLE, SINMAP, TRIGRS, Scoops3D, PLAXIS, and GeoSutio [6,13–17].
2.  Simulate the movement of soil on the slope after landslide only models, such as Flo2D, Landslide2D, PFC, and DDA [6,18,19].
3.  Simultaneously simulate the occurrence of landslide and the movement of soil on the slope after landslide models, such as FLAC, ABAQUS, and Anura [6,20].

In this study, the most important issue is whether the slope is stable or not, which is the reason why limit equilibrium theory is used, but not numerical simulations of landslides, for the selection of parameters for statistical analysis.

### 2.2. Monitoring-Related Research

Slope monitoring data is the basis for the LSL research and early warning predictions. Slope monitoring can be carried out through on-site monitoring and remote sensing.

1.  On-site monitoring

The purpose of on-site monitoring is mainly to obtain on-site data, including surface changes, underground changes, surface hydrology, groundwater hydrology, and structural deformation. The comparison of various on-site monitoring programs is shown in Table 1 [21]. In recent years, new technologies and equipment have been continuously

improved, e.g., GPS [22,23], TDR [23], RGB-D sensors [24,25], and ERT [26]. On-site monitoring has the advantages of high efficiency, high frequency, and accuracy from directly measuring on-site data, and it is quite convenient for subsequent analysis and application. Therefore, the program of on-site monitoring is widely used. However, since the results of on-site monitoring are limited to "points", if wide-range monitoring is to be carried out, a lot of resources have to be invested not only in equipment construction but also in subsequent maintenance because underground monitoring instruments are vulnerable to damage from ground deformations.

**Table 1.** Comparison table of on-site monitoring programs (modified from [21]).

| Investigation Item | Instruments | | Investigation Objects | Accuracy |
|---|---|---|---|---|
| Surface changes | Surface inclinometer | | Tilting direction and amount of ground surface | 1″ |
| | Surface extensometer | | Fracture displacement and velocity | 0.2 mm |
| | Surface measurement | Optical measuring instruments | Tilting direction and amount of ground surface | 1~10 mm |
| | | GNSS | Displacement of the ground surface | NA |
| | | LiDAR scanner | Terrain 3D variation | NA |
| Underground changes | In-place inclinometer | | Sliding surface position and variation | 5~10″ |
| | Pipe strain gauge | | Sliding surface position and variation | $1 \times 10^{-6}$ |
| | Borehole extensometer | | Sliding surface dislocation rate | 0.2 mm |
| | Multipoint borehole extensometer | | Sliding surface position and dislocation rate | 0.3 mm |
| Surface hydrology | Rain gauge | | Rainfall amount | 0.5 mm |
| Underground hydrology | Water level gauge | | Variation of water level in the hole | 0.05%FS |
| | Pore pressure gauge | | Variation of water pressure of the sliding surface | 0.05%FS |
| | Soil moisture meter | | Variation of soil saturation | NA |
| | Flowmeter | | Variation of discharge | NA |
| Structures | Earth pressure gauge | | Earth pressure acting on retaining walls, deep foundation piles | 0.1%FS |
| | Load cell | | Tension acting on the ground anchor | 0.1%FS |
| | Strain gauge | | Deformation of the structure | $1 \times 10^{-6}$ |
| | Rebar gauge | | Stress acting on the rebar gauge | 0.1%FS |
| | Inclinometer | | Tilt variation of structure | 1~10″ |
| | In-place inclinometer | | Bending deformation of steel pipe piles | 5~10″ |

Note: FS—Factor of Safety; ″ —inch; NA—Not Available.

Some of the occurrence times of LSLs in this study were obtained from analyses using the seismic data from the BATS. Since 1992, 42 stations have been set up in Taiwan and offshore islands. Each station is equipped with a broadband seismograph, which can record a wide frequency range of fluctuations due to the characteristics of its sensitive sensors, which can record rich and high-quality seismic waveforms. It can effectively make up for the insufficiency of seismic wave information recorded by acceleration type or traditional narrow-band velocity type seismographs, thereby improving the evaluation accuracy of earthquake location and scale [27].

2. Remote Sensing

Remote sensing mainly includes optical imagery, airborne LiDAR, and radar data. Optical imagery can be obtained from unmanned aerial vehicle (UAV) imagery [28], aerial photos [29], satellite imagery [29–31], etc., and usually requires special processes [30–32] to derive surface variation or trend information. The airborne LiDAR acquires a large amount of point cloud data through intensive light wave scanning ranging, and can derive high-precision surface deformation after data calculation and spatial matching [33–37]. In recent

years, with the gradual miniaturization of LiDAR products, there have been small LiDAR applications that can be matched with UAVs [38]. Radar data analysis evaluates surface variability through radar wave phase changes. In the application of surface variability, the current common applications include differential InSAR (D-InSAR) [39,40], persistent scatterer InSAR (PS-InSAR) [40–48], short baseline subset InSAR (SBAS-InSAR) [49], and temporally coherent point InSAR (TCP-InSAR) [50,51]. The advantage of landslide monitoring through remote sensing data analysis is to obtain the "plane" data of a landslide. As acquisitions of data have to be coordinated with the schedule of the remote sensing payload, and most of the original data have to be processed by some complex interpretation and analysis, the time frame of data acquisition is slower than for on-site monitoring.

Rainfall data for statistical analysis in this study come from the CWB's on-site monitoring data and radar evaluation data, and all the terrestrial data of LSLs are based on airborne LiDAR and satellite optical imagery.

### 2.3. Early Warning-Related Research

By comparing the literature of current LSL early warning studies from Taiwan [52], the United States [53], Japan [54], Italy [55], and Canada [56], it is clear that each country has a different approach to establishing warning systems according to the risks to be faced and the technologies to be mastered. The approaches are follows:

1. Early warning indicators

Through the aforementioned on-site monitoring, remote sensing, and other solutions, on-site information is obtained to establish early warning indicators, including:

- Surface or underground deformation, velocity, or amount of deformation [21,30,40,42,44,49–51];
- variation speed of groundwater level or water level changes [21,57–60];
- rainfall amount [59,61];
- comprehensive indicators [59].

2. Early warning management values

The management values of early warning indicators are determined by statistical methods, which are generally divided into warning management values and evacuation management values [62–68].

3. Real-time simulation

In addition, early warning indicators and management values are also introduced into the numerical model of LSLs, and real-time simulation is performed to provide guidance for warning and evacuation.

## 3. Materials and Methods

The process with major data analysis steps is as follows: case collection and screening (Section 3.1), case confirmation (Section 3.2), occurrence time confirmation (Section 3.3), triggering rainfall analysis (Section 3.4), linear regression analysis (Section 3.5), and nonlinear regression analysis (Section 3.6). Details are delineated in the following subsections.

### 3.1. Case Collection and Screening

The FORMOSAT-2 satellite images of landslides caused by typhoons and torrential rain events from 2004 to 2016 were collected. The images were compared with each other to confirm each landslide type, location, and size. Based on the NCDR's definition of an LSL, 43,519 collected landslide events were screened out and narrowed down to 259 landslides. In addition, new landslides can be classified into the newborn landslide and the expanded landslide (as shown in Figure 1), and 107 newborn landslides were selected for analysis.

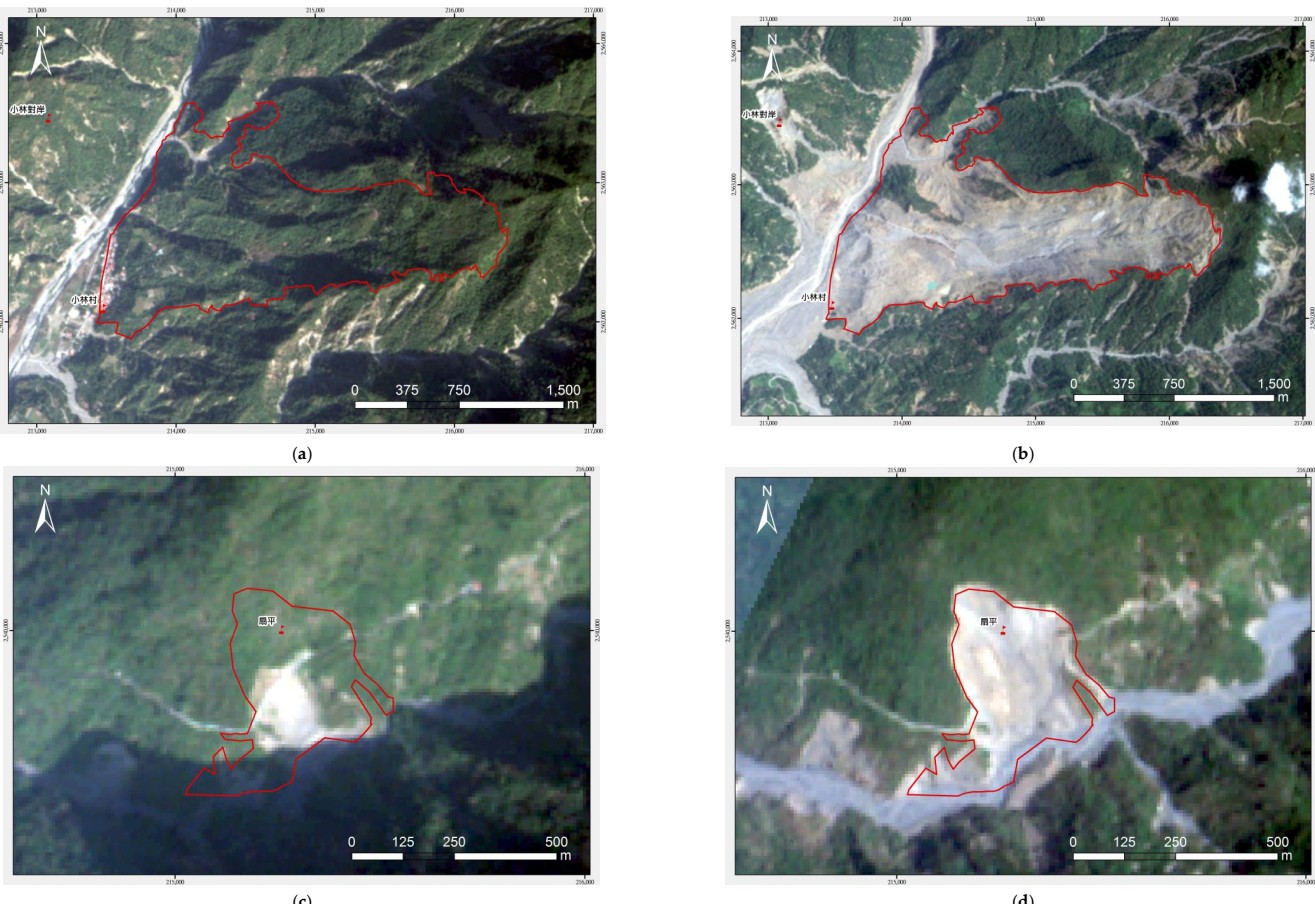

**Figure 1.** Landslide types of satellite images: newborn landslide (**a**,**b**) and expanded landslide (**c**,**d**). (modified from: [66]). (**a**) Xiaolin Village Landslide (pre-event), (**b**) Xiaolin Village Landslide (post-event), (**c**) Shanping Landslide (pre-event), (**d**) Shanping Landslide (post-event).

*3.2. Case Confirmation, Area Size, and Average Slope Identification of LSL*

In this study, the practice of the Central Geological Survey, MOEA on the identification of landslides was referred to [69,70], and landslide cases caused by the influence of specific events were obtained through the creation and comparison of landslide catalogues. The LSLs discussed in this study are mainly new landslides. A total of 259 new LSL cases were identified, including 107 newborn landslides and 152 expanded landslides. Only the 107 newborn landslides were used for analysis. In this stage, area size and average slope of LSLs were obtained at the same time. The projected area $A$ (m$^2$) of an LSL was calculated by the Calculate Geometry function of ESRI ArcMap® software. Based on 5 m resolution DEM data, the slope of each data grid was calculated first with the Slope function of the Surface tool in the Spatial Analyst Tools module of ArcToolbox, and the average slope $\theta$ (degree) for the corresponding range of each newborn LSL was calculated by the Zonal Statistics function in the Zonal tool module.

*3.3. The Occurrence Time Confirmation of LSL*

In order to accurately correlate the triggering rainfall with LSLs, the first task is to confirm the occurrence time of each landslide. The occurrence time data cited in this study mainly come from interviews with local residents [71,72], and the evaluation results of the LSL occurrence time from the BATS [73]. Among the 107 newborn LSL cases, 28 cases were identified with their occurrence time, and for the remaining 79 cases the exact time of occurrence was unknown. The information of the 28 newborn LSLs is shown in Table 2.

**Table 2.** Information of the 28 newborn LSLs.

| No. | ID | Event | Area Size (ha) | Occurrence Time | Cited From |
|---|---|---|---|---|---|
| 1 | SR-3 | | 19 | 09 August 2009 17:00 | |
| 2 | SR-5 | | 238 | 09 August 2009 17:00 | |
| 3 | SR-6 | | 142 | 10 August 2009 12:00 | |
| 4 | SR-7 | | 130 | 09 August 2009 02:00 | |
| 5 | SR-8 | | 88 | 09 August 2009 02:00 | |
| 6 | SR-9 | | 74 | 09 August 2009 04:00 | |
| 7 | SR-11 | | 40 | 08 August 2009 16:00 | |
| 8 | SR-12 | | 32 | 09 August 2009 07:00 | |
| 9 | SR-16 | | 26 | 09 August 2009 07:00 | |
| 10 | SR-19 | | 23 | 08 August 2009 15:00 | Interviews |
| 11 | SR-42 | Typhoon Morakot (200908) | 15 | 09 August 2009 07:00 | with |
| 12 | SR-43 | | 15 | 09 August 2009 07:00 | local |
| 13 | SR-46 | | 15 | 09 August 2009 05:00 | residents |
| 14 | SR-53 | | 14 | 09 August 2009 00:00 | |
| 15 | SR-94 | | 351 | 09 August 2009 10:00 | |
| 16 | SR-95 | | 249 | 09 August 2009 06:00 | |
| 17 | SR-96 | | 81 | 09 August 2009 10:00 | |
| 18 | SR-97 | | 61 | 09 August 2009 09:00 | |
| 19 | SR-98 | | 52 | 09 August 2009 06:00 | |
| 20 | SR-99 | | 15 | 09 August 2009 04:00 | |
| 21 | SR-100 | | 11 | 08 August 2009 10:00 | |
| 22 | SR-101 | | 10 | 09 August 2009 09:00 | |
| 23 | 2005_002 | Typhoon Haitang (200505) | 18 | 21 July 2005 14:33 | The evaluation results of the LSL occurrence time and location from the BATS |
| 24 | 2006_002 | 0609 Torrential Rain | 12 | 10 Jun 2006 00:53 | |
| 25 | 2008_002 | Typhoon Sinlaku (200813) | 89 | 18 September 2008 02:50 | |
| 26 | 2008_003 | Typhoon Kamaegi (200807) | 10 | 19 July 2008 05:30 | |
| 27 | 2012_002 | Typhoon Saola (201209) | 19 | 03 August 2012 09:02 | |
| 28 | 2012_004 | | 25 | 03 August 2012 03:00 | |

*3.4. The Triggering Rainfall Analysis*

From previous methods of analyzing the triggering rainfall of landslides and debris flows [62,68,74,75], the triggering rainfall of LSLs can be described as Equation (1). The rain field can be divided as in Figure 2. The analysis of landslide triggering rainfall *R* should consider $R_0$ and *P*. The cumulative rainfall $R_0$ directly contributes to the landslide event from the beginning of rainfall to the moment the LSL occurred and is called "direct rainfall". The rainfall P is called "antecedent rainfall" that occurred before the start of the current rain event, and is related to the moisture content of soil which also affects the likelihood of having a landslide. The length of time considered for the antecedent rainfall can be adjusted according to the local geological characteristics. In this study, the rainfall of the antecedent seven days is used. The sum of the direct rainfall of the current rainfall event and the antecedent rainfall is the effective cumulative rainfall of the landslide, and is called the triggering rainfall, which can be expressed by Equation (1).

$$R = R_0 + P \approx R_0 + \sum_{i=1}^{N} \alpha^i R_i,$$ (1)

where:

*R* is the triggering rainfall for LSL (mm);

$R_0$ is the accumulated rainfall from the beginning of the rainfall event that caused the LSL to the moment the landslide occurred (mm);
$P$ is the antecedent rainfall (mm) $\approx \sum_{i=1}^{N} \alpha^i R_i$;
$R_i$ is the rainfall on the i-th day (24 h) before the start of the rain field $t_0$ (mm);
$N$ is the number of days to consider the antecedent rainfall (), generally N = 7;
$\alpha$ is the daily (24 h) rainfall triggering landslide decay coefficient (), which can be 0.7 or 0.8. At present, $\alpha$ = 0.7 is used in this study to calculate the antecedent rainfall [62].

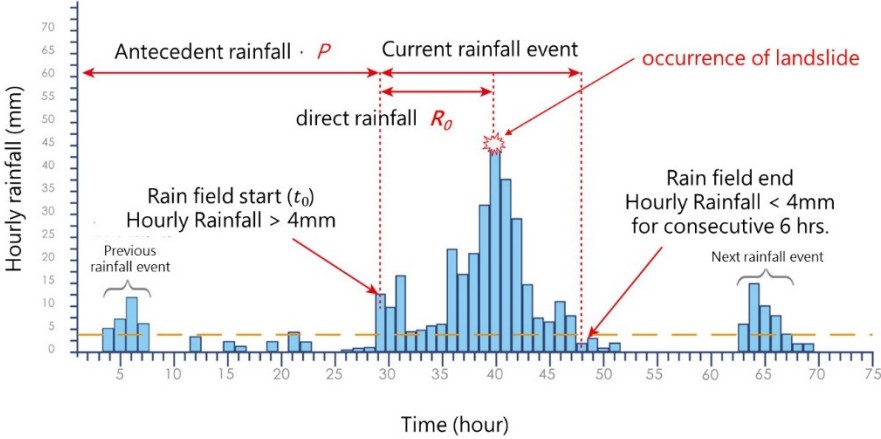

**Figure 2.** Schematic diagram of rain field cutting (modified from: [76]).

The aforementioned 107 newborn LSL cases and the data of 149 potential LSL areas evaluated by the SWCB were used in this study for the follow-up analysis, with the information of the cases shown in Table 3. Among the 107 newborn LSL cases, the occurrence time of 28 cases was confirmed through local resident interviews and evaluation of the data from the BATS, and then triggering rainfall was also calculated. Since the time of occurrence of the remaining 79 cases could not be confirmed, the total rainfall of the corresponding event was used instead of the triggering rainfall. For the 149 potential LSL areas without landslide, the total rainfall of Typhoon Morakot was used as the triggering rainfall in this study. In summary, three different sets of LSL events were used in this study: 28 occurrence-time-known cases, 79 occurrence-time-unknown cases, and 149 potential LSL areas. As for other relevant parameters, the relevant data were collected and estimated based on the actual collapse area and the potential collapse range delineated by the SWCB.

**Table 3.** Information of selected cases.

|  | Cases | Landslide or Not | Rainfall Type Used |
| --- | --- | --- | --- |
| 107 newborn LSLs | 28 occurrence-time-known cases | Yes | Triggering rainfall |
|  | 79 occurrence-time-unknown cases | Yes | Total event rainfall |
| 149 potential LSL areas | | No | Total rainfall of Typhoon Morakot |

*3.5. The Linear Regression Analysis*

When the rain infiltrates the sliding surface and starts to accumulate, the pore water pressure rises. It is assumed that the soil will start to slide when the pore water pressure rises to a critical $h_c$ (m). According to the infinite slope theory, $\theta$ (degree) is defined as the average slope of the sliding surface, $D$ (m) is defined as the thickness of the sliding soil layer, and $h_c$ is defined as landslide triggering pore water pressure or critical pore water pressure for soil sliding. According to Towhata et al. [77], the $h_c/D$ is a function as shown in Equation (2).

$$\frac{h_c}{D} = f\left(\frac{internal\ friction\ angle}{slope\ of\ sliding\ surface}\right) = f\left(\frac{internal\ friction\ angle}{\theta}\right) \qquad (2)$$

Alternatively, $h_c$ is replaced with the landslide triggering rainfall $R$, and the internal friction angle is replaced by the equivalent friction angle $\phi$ (degree). Therefore, Equation (2) can be rearranged in two dimensionless quantities, $R/D$ and $\phi/\theta$, for a linear statistical regression analysis as the following:

$$\left(\frac{R}{D}\right) = a \times \left(\frac{\phi}{\theta}\right) + b. \tag{3}$$

In Equation (3), $R$ can be derived from Equation (1), $\theta$ can be obtained by calculating the average slope by the steps in Section 3.2, and $D$ is the landslide volume $V$ (m³) divided by the projected landslide area $A$ as Equation (4). With an empirical volume–area relation [78] as Equation (4) from the SWCB, $D$ can be evaluated as Equation (5).

$$V = 0.1025 \times A^{1.401} \tag{4}$$

$$D = \frac{V}{A} = 0.1025 \times A^{0.401} \tag{5}$$

Scheidegger mentioned that the equivalent friction coefficient $f$ [] is a function of the landslide volume $V$ as in Equation (6) [79]. Since the friction coefficient $f$ is equal to $\tan \phi$ based on the force balance (gravity and friction) of an incline plane, the equivalent friction angle $\phi$ can be calculated by Equation (7).

$$\log_{10} f = -0.1466 \log_{10} V + 0.62419 \tag{6}$$

$$\phi = \frac{\tan^{-1} f \times 180}{\pi} = \frac{\tan^{-1}\left(e^{-0.1466 \log_{10} V + 0.62419}\right) \times 180}{\pi} \tag{7}$$

### 3.6. The Nonlinear Regression Analysis

To evaluate of the stability of the slope using the concept of FS in Section 2.1, $D$, $R$, $\theta$, and $\phi$ are the four factors to be considered, and a generic nonlinear relationship is assumed as shown in Equation (8).

$$D = f(R, \theta, \phi; a, b, c, d, e) = \frac{a \times R^b \times \theta^c}{\phi^d} + e, \tag{8}$$

where:
$D$ is a nonlinear function of $R, \theta, \phi$ (m),
$a, b, c, d, e$ are the regression coefficients (). To be dimensionally correct, these coefficients should have certain physical units. However, they are just regressionally fitted values and the actual units are ignored in this paper for simplicity.

To deal with this inverse problem, a nonlinear regression is performed with bootstrap resampling to evaluate parameter uncertainty. The analysis steps are shown in Figure 3. The key steps of the entire nonlinear process are detailed as follows.

Step 1. Initial calculation

In this step, the coefficients of $a, b, c, d,$ and $e$ are initially set to be 1.0, and the initial predicted values can be calculated as Equation (8). The (relative) residual is defined as Equation (9).

$$\Delta_i \equiv 1 - \frac{p_i}{o_i}, i = 1 \cdots n, \tag{9}$$

where:

$\Delta_i$ is the residual of the $i$-th data [],
$p_i$ is the landslide thickness of the $i$-th prediction (m),
$o_i$ is the landslide thickness of the $i$-th observation (m),
$n$ is the total number of observations ().

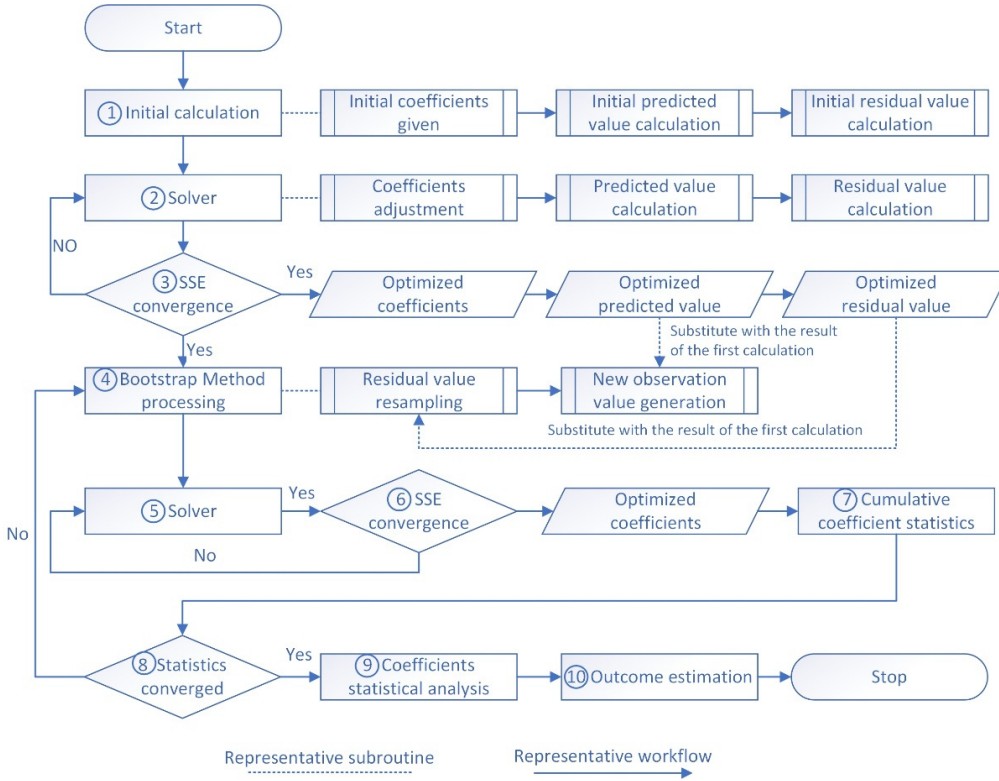

**Figure 3.** Nonlinear regression workflow. The numbered items are the key steps of the process, detailed in the following text.

Step 2. Solver

For the nonlinear regression analysis, the built-in Solver add-in of Microsoft Excel®
is used to find the optimized coefficients $a, b, c, d, e$. The sum of squared errors (SSE) as
in Equation (10) is used as the target function, and the coefficients can be obtained by
minimizing the SSE with Solver. Solver uses the GRG nonlinear solving method to solve
the problem with an accuracy of the constraint of 0.000001.

$$SSE = \sum_{i=1}^{n} \Delta_i{}^2 = \sum_{i=1}^{n} \left( 1 - \frac{p_i}{o_i} \right)^2 \tag{10}$$

Step 3. SSE convergence

To obtain the optimal solution, a convergence criterion of 0.0001 is set in Solver. When
the absolute value of the change of the SSE in the last 5 iterations is less than the convergence
criteria, the GRG nonlinear solution method will stop. With the final optimized coefficient
$\widetilde{a}$, $\widetilde{b}$, $\widetilde{c}$, $\widetilde{d}$, $\widetilde{e}$, the optimized prediction value $\widetilde{p}_i$ and the corresponding optimized residual
value $\widetilde{\Delta}_i$ will be used for the following bootstrap procedure (steps 4 to 8).

Step 4. Bootstrap method processing

The bootstrap method was proposed by Efron in 1979 [80] and can be used to estimate
the uncertainty of system parameters. The concept of this method is to use the existing
data to generate a large number of "phantom samples" through a procedure of resampling
with replacement [81]. Since this method is a nonparametric method, there is no need to
make assumptions about the data distribution pattern [82].

When carrying out the bootstrap method, the optimal set of residual values $\widetilde{\Delta}_i$ from
step 3 is resampled with replacement to obtain a new set of residual value $\Delta_i^*$. With the
rearrangement of Equation (9), the observation value can be described as Equation (11).

Therefore, with the optimized prediction value $\widetilde{p}_i$ and the bootstrapped residual value $\Delta_i^*$, the new observation value $o_i^*$ can be obtained as Equation (12).

$$\Delta_i = 1 - \frac{p_i}{o_i} \Rightarrow o_i = \frac{p_i}{1 - \Delta_i}, \quad i = 1 \cdots n \tag{11}$$

$$o_i^* = \frac{\widetilde{p}_i}{1 - \Delta_i^*}, \quad i = 1 \cdots n \tag{12}$$

Step 5.  Solver

This step is essentially the same as step 2, and the main difference is that the new observation value $o_i^*$ from step 4 will be used for optimization.

Step 6.  SSE convergence

This step is the same as step 3. Different SSEs are solved repeatedly through iterative optimization, and new optimized coefficients $\widetilde{a}^j$, $\widetilde{b}^j$, $\widetilde{c}^j$, $\widetilde{d}^j$, $\widetilde{e}^j$ are obtained for $j$-th calculation of the bootstrap method.

Step 7.  Cumulative coefficient statistics

In this step, the cumulative coefficient of variation (CV) for each optimized coefficient is defined as Equation (13).

$$CV_j = \frac{\sigma_{1\sim j}}{\mu_{1\sim j}}, \quad j = 1 \cdots, \tag{13}$$

where:

$CV_j$ is coefficient of variation of the specific coefficient from the 1st to the $j$-th bootstrap resampling [],

$\sigma_{1\sim j}$ is the standard deviation of the specific coefficient from the 1st to the $j$-th bootstrap resampling [],

$\mu_{1\sim j}$ is the average value of the specific coefficient from the 1st to the $j$-th bootstrap resampling [].

Step 8.  Statistics converged

The variation value $\Delta CV$ is defined as the absolute value of the difference between two consecutive CV values shown in Equation (14) and it can be used to gauge the trend asymptotically stable solution. The convergence criterion for $\Delta CV$ is set to be 0.0001 and steps 4 to 8 are repeated $m$ times until the $\Delta CV$ is converged. In the bootstrap procedure, $m$ is the total times of resampling and is not a preset number but can only be determined by the convergence in this step.

$$\Delta CV_j = \left| CV_j - CV_{j-1} \right|, \quad j = 2 \cdots m \tag{14}$$

Step 9.  Coefficient statistical analysis

After the iterative outcomes converge to stability, the statistical analysis of $\widetilde{a}^j$, $\widetilde{b}^j$, $\widetilde{c}^j$, $\widetilde{d}^j$, $\widetilde{e}^j$ is performed to obtain the maximum value, minimum value, average value, mode value, and statistical values of 40% and 60% confidence intervals of each coefficient.

Step 10. Outcome estimation

With the nonlinear relationship of $D$ from the previous step, the new predicted $D$ can be calculated with the existing LSL data. By comparing the calculated values with the existing data, the prediction ability of each nonlinear relationship of $D$ can be estimated.

## 4. Results

Through the data preparation steps in Sections 3.1–3.3, all the data were analyzed according to Sections 3.4 and 3.5, and the results are described in the following sections.

### 4.1. Linear Regression Analysis Results

With a linear relationship between dimensionless ($R/D$) and dimensionless ($\phi/\theta$) according to Equation (3), the regression results are illustrated as Figure 4. In the figure, the black dots are the 28 known occurrence time cases, the red line is the regression trend line of Equation (15), and the green and purple dotted lines are the upper and the lower boundary lines of Equations (16) and (17), respectively.

$$\text{Trend line}: \quad \frac{R}{D} = 0.1347 \times \frac{\phi}{\theta} - 0.0281, \quad R^2 = 0.55 \tag{15}$$

$$\text{Upper boundary line}: \quad \frac{R}{D} = 0.1347 \times \frac{\phi}{\theta} + 0.0032 \tag{16}$$

$$\text{Lower boundary line}: \quad \frac{R}{D} = 0.1347 \times \frac{\phi}{\theta} - 0.0543 \tag{17}$$

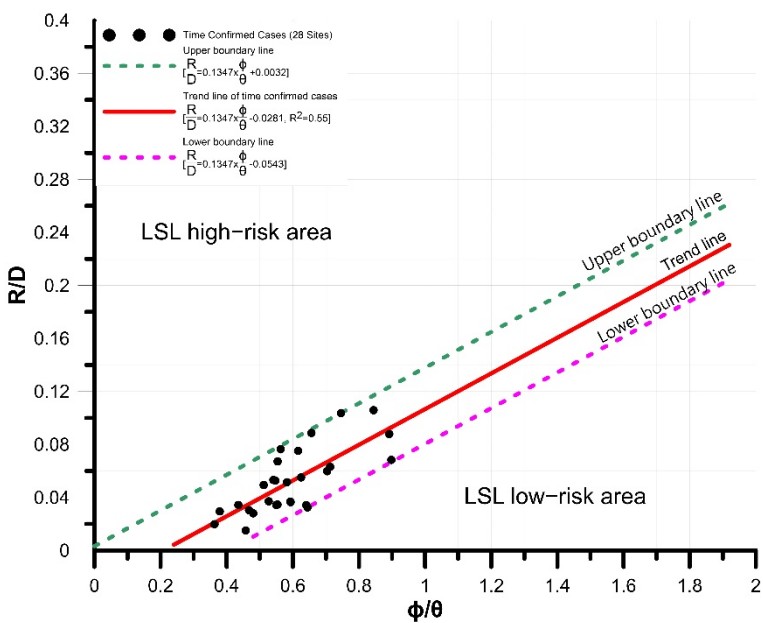

**Figure 4.** Dimensionless factor relationship of LSL cases with known occurrence time.

In order to examine whether the relationship of Equation (15) is reasonable, this study collected data according to the procedures of Sections 3.1–3.3 for the 79 occurrence-time-unknown cases and 149 potential areas delineated by the Soil and Water Conservation Bureau (SWCB). Since there is no information about the time of landslide for these two types of cases, the effective cumulative rainfall is evaluated alternatively. The total accumulated rainfall values of the corresponding events were used for the 79 occurrence-time-unknown cases, and the maximum total accumulated rainfall recorded over the years was taken for the 149 potential areas. After calculating the two dimensionless quantities, the results were superimposed with the trend line and the boundary lines from Figure 5 to display in Figures 6 and 7.

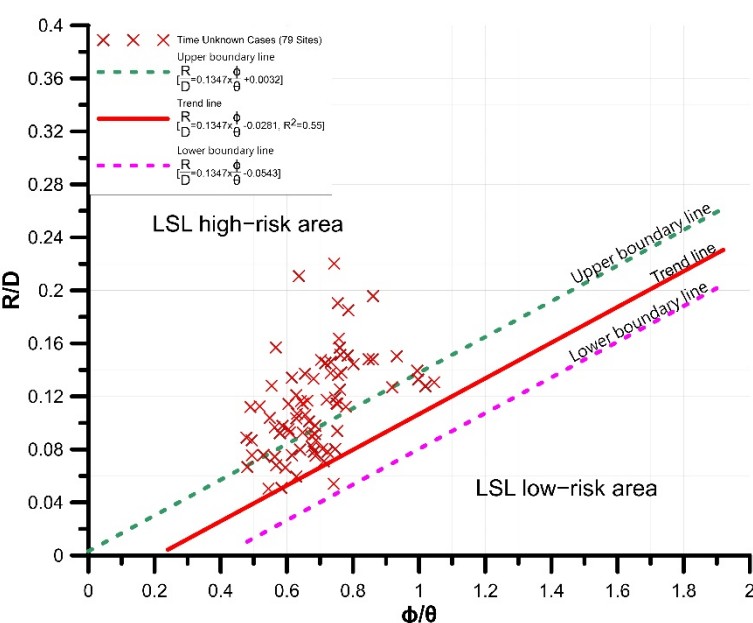

**Figure 5.** Dimensionless factor relationship of LSL cases with unknown occurrence time.

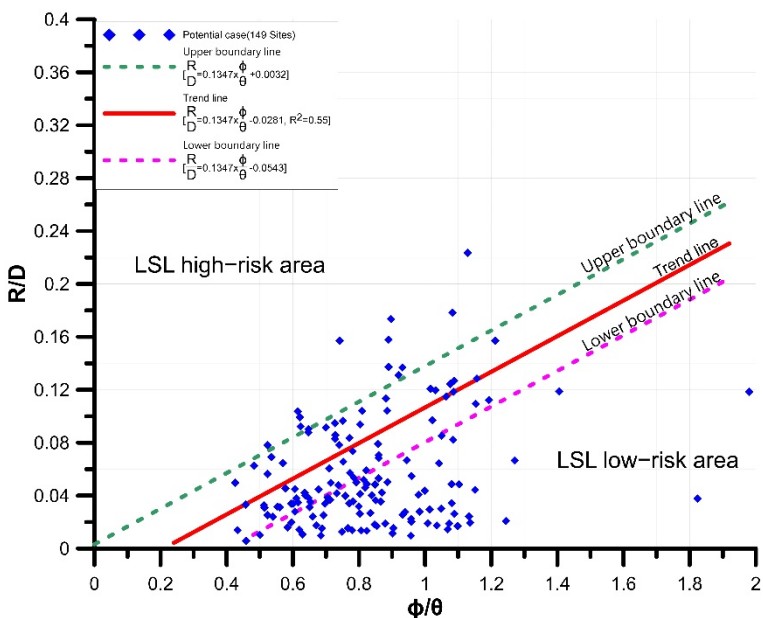

**Figure 6.** Dimensionless factor relationship of potential LSL area.

In Figure 5, almost all cases are above the trend line, and only one case falls between the trend line and the lower boundary line. Since the total rainfall should cap the actual effective cumulative rainfall, it is reasonably expected that most of the data (if not all) are above the trend line.

Among the 149 records in Figure 8, 67 records are located below the lower boundary line (45%), 112 below the trend line (75%), 135 below the upper boundary line (91%), and 14 above the upper boundary line (9%).

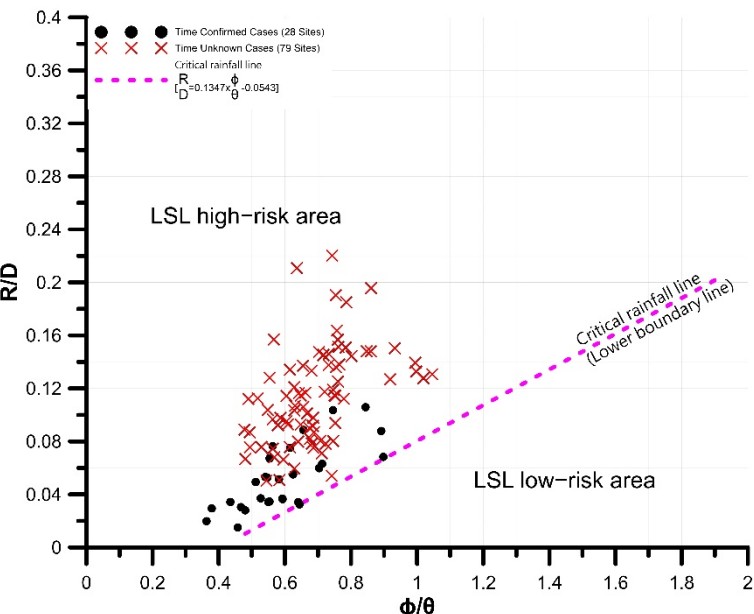

**Figure 7.** Dimensionless factor relationship of potential LSL area.

To be conservative, the lower boundary line (the purple dotted line in Figure 9) of the occurrence rainfall is suggested to be the management value for LSL evacuation and refuge.

### 4.2. Nonlinear Regression Analysis Results

In this study, the Analysis ToolPak of Microsoft Excel® was used to perform the nonlinear fit analysis between the predictions and the observations according to Equation (8). The results are shown in Table 4 and Figure 8. From the value of R square and the adjusted R square, it was found that Equation (8) has a good degree of model fit, and F is very different from the significance F (last two columns of ANOVA). This indicates that each parameter has large differences between groups, and small differences within groups. From the results in Figure 8b, it was found that the predicted value and the observed value have similar trends, and the function described by Equation (8) is reasonable.

**Table 4.** Result of complex data analysis.

| Regression Statistics | | | | | | |
|---|---|---|---|---|---|---|
| Multiple R | | | | | | 0.9569 |
| R Square | | | | | | 0.9157 |
| Adjusted R Square | | | | | | 0.9052 |
| Standard Error | | | | | | 2.8447 |
| Observations | | | | | | 28 |
| **ANOVA** | | | | | | |
| | df | SS | MS | F | | Significance F |
| Regression | 3 | 2109.4427 | 703.1476 | 86.8894 | | $4.9903 \times 10^{-13}$ |
| Residual | 24 | 194.2187 | 8.0924 | | | |
| Total | 27 | 2303.6614 | | | | |
| | Coefficients | Standard Error | *t*-Test | *p*-value | Lower 95% | Upper 95% |
| Intercept | 66.7058 | 6.1858 | 10.7838 | $1.10 \times 10^{-10}$ | 53.9390 | 79.4725 |
| R(m) | −2.2273 | 2.8034 | −0.7945 | 0.4347 | −8.0133 | 3.5587 |
| θ(degree) | 0.0043 | 0.1279 | 0.0334 | 0.9737 | −0.2597 | 0.2682 |
| $\phi$(degree) | −2.2251 | 0.1456 | −15.2793 | $7.26 \times 10^{-14}$ | −2.5257 | −1.9246 |

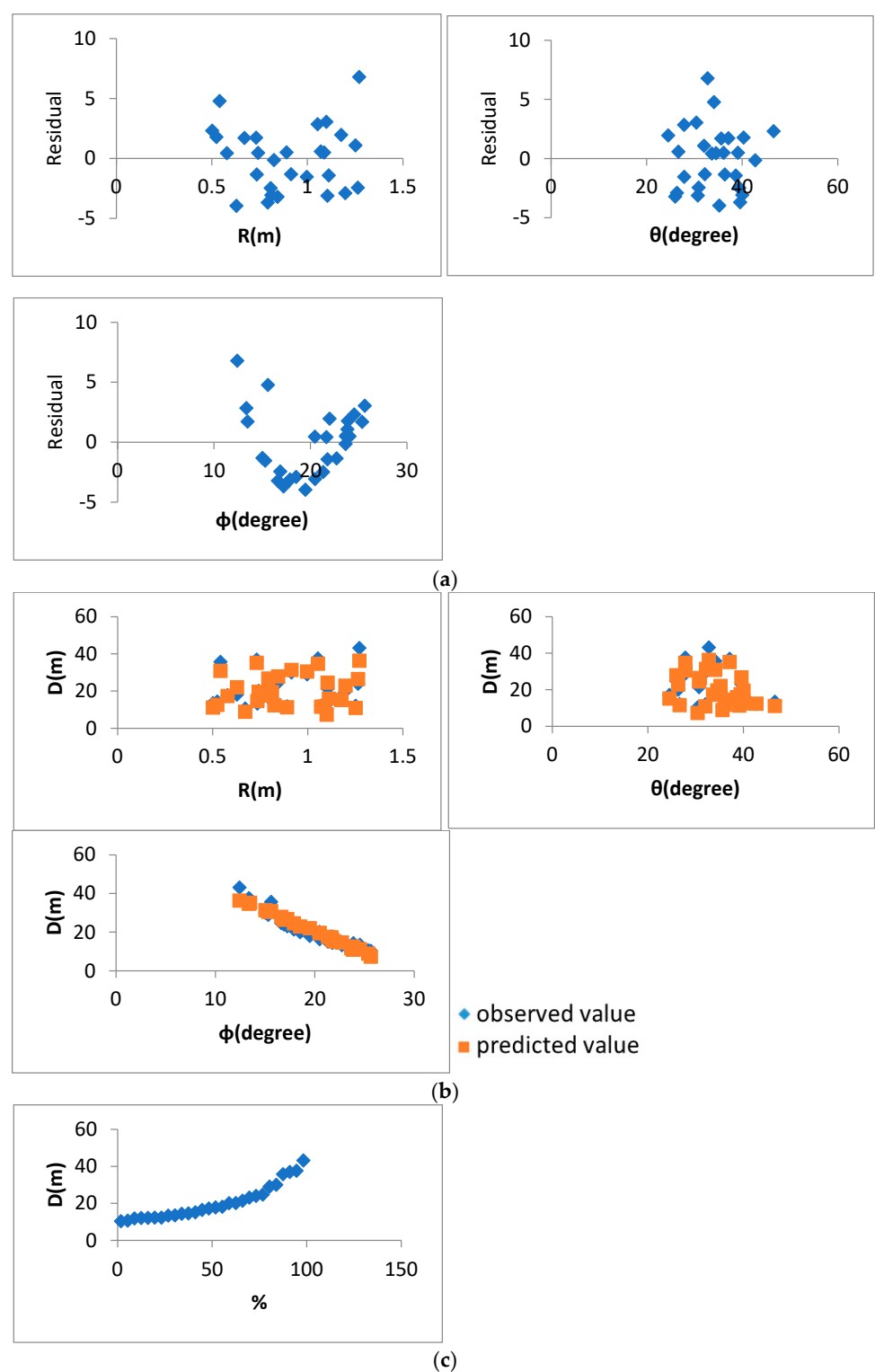

**Figure 8.** The results of the fit analysis between the nonlinear predicted value and the observed value, where (**a**) is the residual of each parameter, (**b**) is the sample regression line of each parameter, and (**c**) is the normal probability. *D* is the thickness of the sliding soil layer, *R* is the landslide triggering rainfall, $\theta$ is the slope of sliding surface, and $\phi$ is the equivalent friction angle.

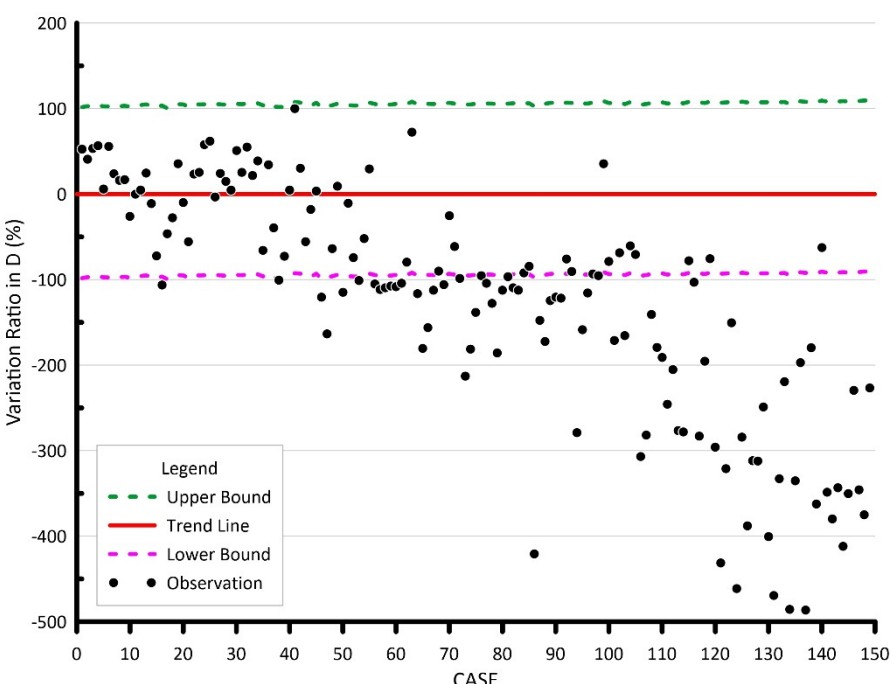

**Figure 9.** Comparison between predicted and estimated values of collapse thickness in 149 potential LSL areas.

After confirming that Equation (8) was reasonable, the nonlinear regression analysis according to the procedures in Figure 5 was performed. With 5000 iterations of calculation, it was found that the variation of the CV of each coefficient is approaching asymptotic stability as the number of iterations increases. The optimized coefficients of each group were obtained from the statistics in Table 5. Through the results in Table 5, the statistical values of each coefficient were substituted into Equation (8) to calculate the predicted value. The predicted values were compared with the observed values of 28 occurrence-time-known cases and 79 occurrence-time-unknown cases. The results of landslide thickness difference $\Delta D$ are shown in Tables 6 and 7.

**Table 5.** Statistics of nonlinear regression coefficients.

| Parameter | Max | Min | Mean | Median | 40% | 60% |
|---|---|---|---|---|---|---|
| a | 24,469.2875 | 648.5690 | 6095.1517 | 4438.0571 | 3576.8579 | 5539.8139 |
| b | −0.0229 | −0.3621 | −0.1647 | −0.1604 | −0.1744 | −0.1460 |
| c | 0.1614 | −0.4220 | −0.1339 | −0.1298 | −0.1492 | −0.1089 |
| d | 2.4898 | 0.8831 | 1.6260 | 1.6288 | 1.5488 | 1.7067 |
| e | 5.0123 | −22.1158 | −4.4903 | −3.5814 | −4.7650 | −2.5826 |

**Table 6.** The landslide thickness of the prediction minus the observation for 28 occurrence-time-known cases. (Unit: m, % relative to the observation).

| | $\Delta D_{max}$ | $\Delta D_{min}$ | $\Delta D_{mean}$ | $\Delta D_{median}$ | $\Delta D_{40\%}$ | $\Delta D_{60\%}$ |
|---|---|---|---|---|---|---|
| Standard deviation | 9.18 (15%) | 6.85 (35%) | 3.52 (9%) | 1.38 (6%) | 1.65 (6%) | 1.77 (7%) |
| Mean of absolute values | 15.03 (72%) | 31.00 (167%) | 8.03 (41%) | 1.07 (5%) | 2.20 (11%) | 2.87 (15%) |
| Maximum value | 42.61 (99%) | −23.20 (−117%) | 16.15 (55%) | 2.38 (12%) | 0.02 (0%) | 6.63 (26%) |
| Mean value | 15.03 (72%) | −31.00 (−167%) | 8.03 (41%) | 0.16 (1%) | −2.20 (−11%) | 2.82 (15%) |
| Median value | 11.38 (76%) | −28.18 (−163%) | 6.94 (43%) | 0.45 (3%) | −1.85 (−10%) | 2.84 (17%) |
| Minimum value | 6.38 (37%) | −50.56 (−231%) | 3.69 (21%) | −4.00 (−13%) | −7.11 (−23%) | −0.36 (−2%) |

**Table 7.** The landslide thickness of the prediction minus the observation for 79 occurrence-time-unknown cases. (Unit: m, % relative to the observation).

|  | $\Delta \mathbf{D_{max}}$ | $\Delta \mathbf{D_{min}}$ | $\Delta \mathbf{D_{mean}}$ | $\Delta \mathbf{D_{median}}$ | $\Delta \mathbf{D_{40\%}}$ | $\Delta \mathbf{D_{60\%}}$ |
|---|---|---|---|---|---|---|
| Standard deviation | 1.42 (4%) | 1.60 (20%) | 1.20 (8%) | 0.65 (5%) | 0.69 (6%) | 0.67 (5%) |
| Mean of absolute values | 9.26 (75%) | 26.47 (216%) | 4.39 (35%) | 0.63 (5%) | 2.13 (17%) | 1.48 (12%) |
| Maximum value | 16.21 (83%) | −22.69 (−150%) | 8.71 (61%) | 1.62 (15%) | 0.13 (1%) | 3.36 (30%) |
| Mean value | 9.26 (75%) | −26.47 (−216%) | 4.39 (35%) | −0.39 (−3%) | −2.13 (−17%) | 1.48 (12%) |
| Median value | 8.81 (75%) | −26.41 (−219%) | 4.19 (35%) | −0.43 (−3%) | −2.17 (−18%) | 1.42 (12%) |
| Minimum value | 7.46 (62%) | −32.25 (−254%) | 1.85 (17%) | −1.72 (−16%) | −3.55 (−32%) | 0.02 (0%) |

From Tables 6 and 7, Equation (18) was obtained by substituting the median value of each coefficient into Equation (8) which has a relatively good predictive ability in both cases with or without known occurrence time. Assuming a normal distribution of Equation (8), the upper and lower boundaries of the 99.7% range covered by three standard deviations were used. The lower boundary illustrated in Equation (19) was obtained by the 40th percentile values of each coefficient, and the upper boundary illustrated in Equation (20) was obtained by the 60th percentile value of each coefficient.

$$\text{Trend line}: \ D_{TL} = \frac{4438.0571 \times R^{-0.1604} \times \theta^{-0.1298}}{\phi^{1.6288}} - 3.5814 \tag{18}$$

$$\text{Upper boundary line}: \ D_{LB} = \frac{3576.8579 \times R^{-0.1744} \times \theta^{-0.1492}}{\phi^{1.5488}} - 4.7650 \tag{19}$$

$$\text{Lower boundary line}: \ D_{UB} = \frac{5539.8139 \times R^{-0.1460} \times \theta^{-0.1089}}{\phi^{1.7067}} - 2.5826 \tag{20}$$

To better illustrate the data distribution for the 149 potential LSL areas, the trend line and the boundary lines were transformed and normalized. The results from Equation (18) were set as the zero-reference line, and those from Equations (19) and (20) were scaled to be +100% (upper boundary) and −100% (lower boundary), respectively. The three lines of the 149 observation data are plotted in Figure 9. Of the 149 records, 83 records (56%) are below the lower boundary line, 115 records (77%) below the trend line, and all 149 records (100%) are below the upper boundary line.

## 5. Discussion

According to the aforementioned information and results, some issues are discussed in the following sections.

### 5.1. Prediction Ability

In order to directly compare the predictive capabilities (rainfall is the key indicator for early warning systems) of the results from linear regression analysis and nonlinear regression analysis, Equations (15) to (20) can be rewritten with $R$ as the dependent variable as Equations (21) to (26).

$$R_{LT} = D \times \left( 0.1347 \times \frac{\phi}{\theta} - 0.0281 \right), \tag{21}$$

$$R_{LU} = D \times \left( 0.1347 \times \frac{\phi}{\theta} + 0.0032 \right), \tag{22}$$

$$R_{LL} = D \times \left( 0.1347 \times \frac{\phi}{\theta} - 0.0543 \right), \tag{23}$$

$$R_{NT} = \left[ \frac{(D + 3.5814) \times \phi^{1.6288}}{4438.0571 \times \theta^{-0.1298}} \right]^{\frac{1}{-0.1604}}, \tag{24}$$

$$R_{NU} = \left[ \frac{(D + 2.5826) \times \phi^{1.7067}}{5539.8139 \times \theta^{-0.1089}} \right]^{\frac{1}{-0.1460}}, \tag{25}$$

$$R_{NL} = \left[ \frac{(D + 4.7650) \times \phi^{1.5488}}{3576.8579 \times \theta^{-0.1492}} \right]^{\frac{1}{-0.1744}}, \tag{26}$$

where:

$R_{LT}$ is the predicted rainfall by the linear regression trend line (m),
$R_{LU}$ is the predicted rainfall by the linear regression upper boundary line (m),
$R_{LL}$ is the predicted rainfall by the linear regression lower boundary line (m),
$R_{NT}$ is the predicted rainfall by the nonlinear regression trend line (m),
$R_{NU}$ is the predicted rainfall by the nonlinear regression upper boundary line (m),
$R_{NL}$ is the predicted rainfall by the nonlinear regression lower boundary line (m).

With the 79 occurrence-time-unknown cases and the 149 potential areas, the predicted rainfalls were calculated by Equations (21) to (26), the trend lines and boundary lines were transformed and scaled to be 0% and ±100%, similar to Figure 9, and the results are shown in Figures 10–13.

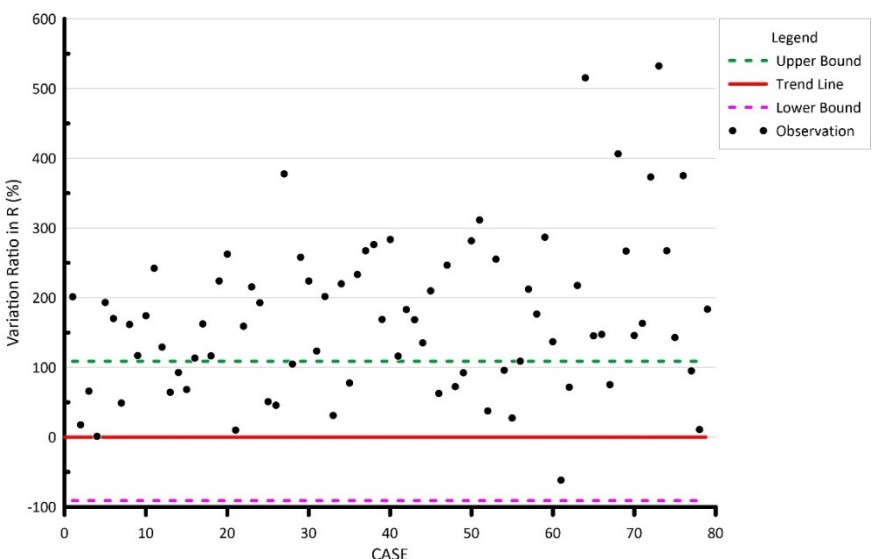

**Figure 10.** Prediction of rainfall distribution by linear analysis of 79 LSL cases.

In Figure 10 (the linear regression analysis results of 79 occurrence-time-unknown cases), all observed rainfall data are greater than the predictive ones, indicating no Type-II error (false negative).

In Figure 11, there are Type-I errors (false positives). Among the 149 cases, 12 (about 8%) are above the upper boundary of the predictive rainfall but no landslide occurred. For the results of the nonlinear regression analysis in Figure 12, there is one case (about 1%) below the lower boundary of the predictive rainfall, which is a Type-II error. According to the results in Figure 13, all of the 149 cases are below the upper boundary of the predictive rainfall, which means no Type-I error.

In summary, based on the limited data used in this study, there is no Type-II error from linear regression and no Type-I error from nonlinear regression. If different situations occur in the future, the new data should be updated and included for improving risk probability assessment.

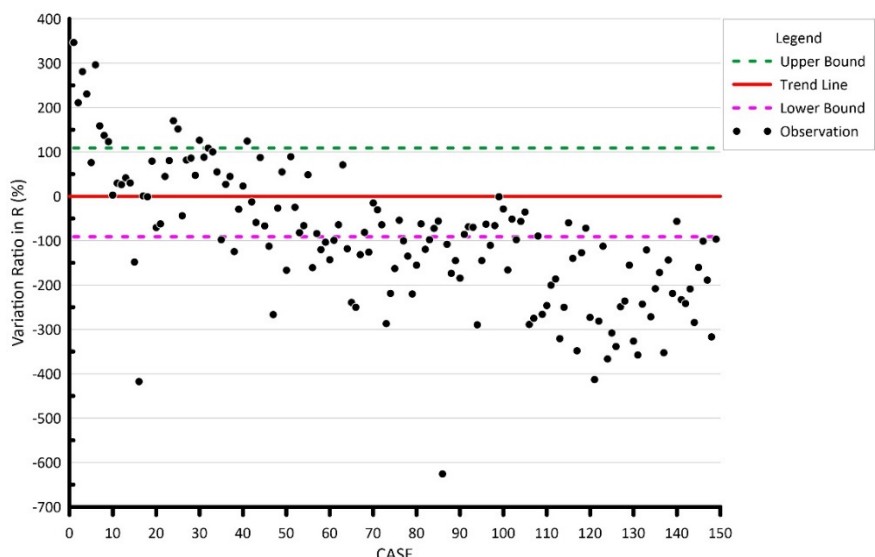

**Figure 11.** Prediction of rainfall distribution by linear analysis of 149 potential LSL areas.

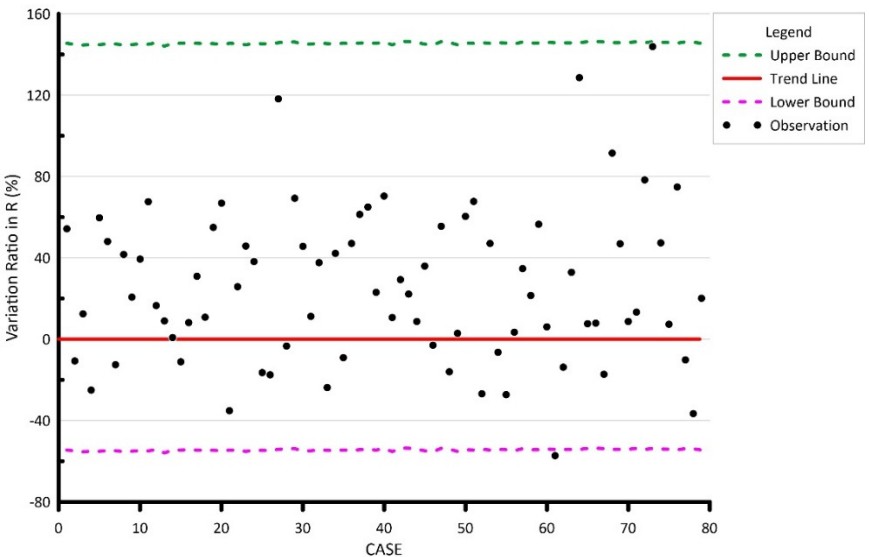

**Figure 12.** Prediction of rainfall distribution by nonlinear analysis of 79 LSL cases.

With Equations (21) to (26) for the 28 occurrence-time-known cases, the predicted rainfalls were compared with the observed triggering rainfalls, and error sums of squares and the root mean square errors are shown in Table 8. By examining the error sum results, the nonlinear trend ($R_{NT}$) has the best predictive capacity. Following the same procedures, 79 occurrence-time-unknown cases and 149 potential areas were evaluated and the results are shown in Tables 9 and 10. It is also found that the nonlinear trend line has the best predictive capacity. Nevertheless, the conservative lower boundary value should be used regarding the evacuation management value of the early warning operation. From the comparison of the error sums in Tables 8 and 9, the estimates by the lower boundary line of nonlinear analysis do provide better predictions.

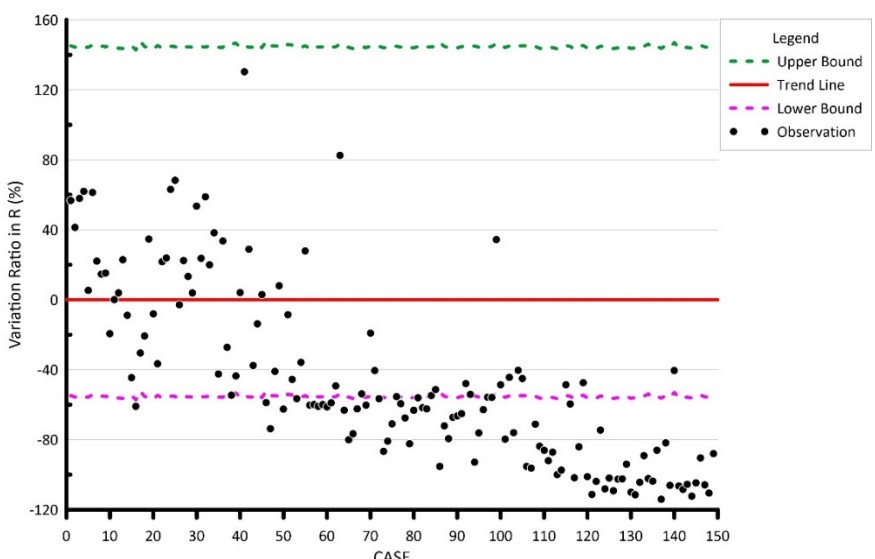

**Figure 13.** Prediction of rainfall distribution by nonlinear analysis of 149 LSL potential areas.

**Table 8.** Statistics of rainfall predictions for 28 occurrence-time-known LSL cases.

| | $R$ | $R_{LT}$ | $R_{LL}$ | $R_{LU}$ | $R_{NT}$ | $R_{NL}$ | $R_{NU}$ |
|---|---|---|---|---|---|---|---|
| Maximum value (mm) | 1271.3 | 1599.1 | 1147.8 | 2554.0 | 1425.8 | 804.3 | 3069.4 |
| Minimum value (mm) | 501.6 | 571.4 | −198.5 | 955.9 | 277.0 | 178.7 | 507.4 |
| Error sum of squares (m$^2$) | | 2.3922 | 9.0971 | 19.3008 | 2.1096 | 4.3647 | 55.0174 |
| Normalized error sum of squares | | 4.1646 | 9.9850 | 34.2229 | 3.2921 | 4.8297 | 70.8433 |
| Standard deviation (mm) | | 297.7 | 580.5 | 845.5 | 279.5 | 402.1 | 1427.5 |
| Normalized standard deviation | | 0.3927 | 0.6081 | 1.1258 | 0.3492 | 0.4229 | 1.6198 |

**Table 9.** Statistics of rainfall predictions for 79 occurrence-time-unknown LSL cases.

| | $R$ | $R_{LT}$ | $R_{LL}$ | $R_{LU}$ | $R_{NT}$ | $R_{NL}$ | $R_{NU}$ |
|---|---|---|---|---|---|---|---|
| Maximum value (mm) | 1994.4 | 1544.9 | 1174.1 | 1987.9 | 1688.7 | 950.8 | 3589.9 |
| Minimum value (mm) | 779.2 | 464.8 | 132.2 | 862.1 | 872.7 | 470.0 | 1966.4 |
| Error sum of squares (m$^2$) | | 11.2251 | 26.5031 | 3.7198 | 4.1568 | 18.3132 | 40.8837 |
| Normalized error sum of squares | | 4.5775 | 12.7829 | 1.9193 | 1.5839 | 7.6505 | 31.4102 |
| Standard deviation (mm) | | 644.8 | 990.8 | 371.2 | 392.4 | 823.6 | 1230.5 |
| Normalized standard deviation | | 0.4117 | 0.6881 | 0.2666 | 0.2422 | 0.5323 | 1.0786 |

**Table 10.** Statistics of rainfall predictions for 149 potential LSL areas.

| | $R$ | $R_{LT}$ | $R_{LL}$ | $R_{LU}$ | $R_{NT}$ | $R_{NL}$ | $R_{NU}$ |
|---|---|---|---|---|---|---|---|
| Maximum value (mm) | 2267.9 | 3016.4 | 2685.2 | 3412.0 | 2739.7 | 1575.6 | 5647.8 |
| Minimum value (mm) | 822.3 | 804.5 | 511.1 | 1037.4 | 1144.4 | 591.5 | 2583.9 |
| Error sum of squares (m$^2$) | | 10.0854 | 20.0943 | 8.5405 | 6.1552 | 23.6371 | 94.8223 |
| Normalized error sum of squares | | 3.1882 | 5.8767 | 3.4506 | 2.3811 | 6.6163 | 44.9751 |
| Standard deviation (mm) | | 611.2 | 862.7 | 562.4 | 477.5 | 935.7 | 1874.0 |
| Normalized standard deviation | | 0.3436 | 0.4665 | 0.3575 | 0.2970 | 0.4950 | 1.2906 |

*5.2. Limitations*

In order to establish the specific relationship between LSLs and triggering rainfall for the future LSL early warning predictions, this study collected LSL case data for modeling. As aforementioned, under the situation that epistemic conditions of LSLs are insufficient, coupled with the fact that LSL is not common, the data that can be collected are very limited. Although the model verified the rationality, it does not mean this model could be used directly in another region outside Taiwan, and those who want to apply this model should follow the procedures mentioned in this study to retrieve the suitable parameters before practice.

*5.3. Early Warning Application*

When the evacuation of early warning systems is announced, it is important to take into account not only the number of evacuees, the length of the evacuation route, and the opening of evacuation shelters, but also the time required for the activation of the evacuation mechanism, agency communication, evacuation notification and enforcement, evacuation status confirmation, etc. It is recommended that the operation should be carried out 3 to 6 h before the cumulative rainfall value reaches the evacuation management value. For this reason, it is suggested to use the quantitative rainfall forecast of 3 to 6 h, such as: ensemble model-based typhoon quantitative precipitation forecast (ETQPF) or blended quantitative precipitation forecast (BQPF). When the forecast of accumulated rainfall reaches the critical value, the evacuation mechanism should be activated immediately for an adequate response time.

## 6. Conclusions

When a large-scale landslide (LSL) disaster occurs, it could have a great impact on people's lives and properties. In response to the potential threat caused by LSLs, this study collected most relevant data of LSL cases, analyzed them to quantify the relationship between the LSL and the triggering rainfall, and proposed to apply this relationship to LSL warning predictions.

The satellite imagery and additional information of landslides from 2004 to 2016 were collected in this study. After screening of 43,519 landslide records, 107 newborn landslides were selected for analysis, including 28 occurrence-time-known cases and 79 occurrence-time-unknown cases. In addition, 149 potential LSL areas evaluated by the Soil and Water Conservation Bureau (SWCB) were also used for improving the lower boundary line.

This study employs two methods of linear and nonlinear regression analysis to assess the relationship between LSL and rainfall. The results show that there are 8% Type-I errors (false positives) in the linear regression analysis, and 1% Type-II errors (false negatives) in the nonlinear regression analysis. With the comparison of statistical indicators, the trend line of nonlinear regression analysis shows better predictive power. Considering the response time required for the early warning operation, it is suggested that the nonlinear lower boundary line can be used as the evacuation and refuge management value. Combined with the numerical rainfall forecast of 3 to 6 h, it can be applied to the evacuation and refuge operations of LSLs.

LSL is not common, but always causes serious impacts. In this study, although a reasonable model for Taiwan was successfully established through limited data, it does not mean that it can be directly applied to other regions. In order to make this model more widely used, testing through foreign cases and expanding the factors considered, such as: geological type, slope aspect, water content, land use, NDVI, time error effect, etc., would be the future topics for further understanding and applications of LSL-related issues.

**Author Contributions:** Conceptualization, C.-L.S., T.-T.T., Y.-J.T. and J.H.-C.W.; methodology, T.-T.T., Y.-J.T., C.-L.S. and J.H.-C.W.; data curation, Y.-J.T.; writing original draft, T.-T.T.; visualization, T.-T.T. and Y.-J.T.; supervision, J.H.-C.W.; project administration, C.-L.S. All authors have read and agreed to the published version of the manuscript.

**Funding:** This research was funded by the Soil and Water Conservation Bureau, Council of Agriculture, Executive Yuan, Taiwan, grant number SWCB-109-153, and the APC was funded by Disaster Prevention Education Center, National Cheng-Kung University.

**Data Availability Statement:** The data that support the findings of this study are available from the author Y.-J.T., upon reasonable request.

**Acknowledgments:** We thank the Soil and Water Conservation Bureau of Council of Agriculture under Executive Yuan for providing information regarding the project, large-scale landslide disaster prevention, and mitigation under climate change.

**Conflicts of Interest:** The authors declare no conflict of interest.

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
