# Peer review of "Triggering Rainfall of Large-Scale Landslides in Taiwan: Statistical Analysis of Satellite Imagery for Early Warning Systems"

_water, doi:10.3390/w14213358_

Round 1
Reviewer 1 Report
Authors presented an approach to the Triggering Rainfall Characteristics of Large-Scale Landslide: from Satellite Imagery Analysis, Statistical Analysis, to Early Warning Applications. The subject is interesting enough to merit publication. However, there are certainly major concerns that I think are reasonable, and could further improve the quality of the manuscript before its final acceptance or publication.
Comments
· The concluding remarks of the abstract are not well-written. It’s merely the repetition of the objectives and title of the manuscript.
· Introduction section lacks a clear research hypothesis. Please stress to provide this clear articulation.
· In some places, the introduction is misleading and not very clear in the direction. The contents sometimes jump between topics without a clear direction.
· Have checked multi-collinearity? If not then stress it.
· Have you validated the model?
· The general description of the problem (introduction section) and its importance in science and for society could be further improved.
· Need a further explanation of the findings instead of only comparing with previous literature. It would be nice to see if the authors establish logical arguments and the actual meanings of the findings.
· I noticed that the conclusion section tends to repeat abstract and results. The conclusion paragraph should be short, impactful, and direct the reader to the next steps and opportunities in this research.
· Do you think many people will be searching for some of these specific words (large-scale landslide, triggering rainfall characteristic, etc.)? Consider some new terms that are novel and enable more folks to find your article.
· Discussion is the weakest section of the draft m/s since it currently lacks: (i) a subsection that clearly articulates your work’s main limitations, and (ii) a subsection clearly articulating the broader applicability of your methods and findings beyond your case study.
Reviewer 2 Report
This manuscript presents a landslide case study that took place in Taiwan. The research herein presented is certainly within the scope of Water.
According to my observations, the topic of the manuscript is interesting and challenging. However, several points need to be addressed prior to its publication. I provide a list of comments below:
Comments:
1. The title needs some careful rephrasing. At this stage is too long and it does not mention that this is about a case study analysis in Taiwan.
2. The abstract needs to introduce the location of the landslide, i.e. the study area. Currently the first three lines are rather general and unprecise.
2. The introduction needs re-arrangement. It cannot contain sub-sections or bullet points. It needs to be a continuous message. If needed, additional sections can be added after the introduction. Please, check other papers published in Water.
3. Line 64, on numerical models. Please, be more specific about those models with regard to the numerical characteristics (2D, 3D, finite volumes?). Also, consider [1] and the importance of mesh topology when dealing with this type of geophysical flows.
4. Line 85, on remote section. The authors explained that InSAR and LiDAR are methods to identify and monitor landslides. Nonetheless, I think the authors should also outline other existing monitoring methods to measure the landslides phenomena, i.e. GPS, borehole inclinometer [2], time-domain reflectometry (TDR) [3], optical fiber sensing technology [3] or RGB-D sensors to measure the surface deformation [5,6]. Only after describing these monitoring methods the authors can state the methods they decided to use.
5. Based on the methodological manipulations explained in the manuscript (section 2), is there any estimate on how the occurrence time error will be amplified based on the methodological manipulations of each variable displayed in section 2.5?
6. Following with my previous comment, how the accuracy in the predictions is related with landslide characteristics (velocity, soil depth, porosity, water content)?. Furthermore, I would welcome a justification/discussion on the existence of distinct land use or NDVI values within the study area.
7. Figure 8. The caption of the figure needs to explain each variable (phi, theta, R).
8. Line 411. The word “In” is duplicated.
9. Conclusions. Recall in the conclusions the meaning of each acronym (LSL, SWCB, etc).
Bibliography
[1] Juez, C., Murillo, J., García-Navarro. 2D simulation of granular flow over irregular steep slopes using global and local coordinates. Journal of Computational Physics, Volume 255, 2013.
[2] Zhang Y, Tang H, Li C, Lu G, Cai Y, Zhang J, Tan F. Design and Testing of a Flexible Inclinometer Probe for Model Tests of Landslide Deep Displacement Measurement. Sensors 2018, 18, 224.
[3] Su, M.-B.; Chen, I.-H.; Liao, C.-H. Using TDR cables and GPS for landslide monitoring in high mountain area. J. Geotech. 669 Geoenviron. Eng. 2009, 135, 1113–1121.
[4] Zhu HH, Shi B, Zhang CC. FBG-Based Monitoring of Geohazards: Current Status and Trends. Sensors 2017, 17, 452.
[5] Caviedes-Voullième, D., et al., 2014. 2D dry granular free-surface flow over complex topography with obstacles. part I: experimental study using a consumer-grade RGB-D sensor. Computers & Geosciences 73 (0), 177–197.
[6] C. Juez, D. Caviedes-Voullième, J. Murillo and P. García-Navarro. 2D dry granular free-surface transient flow over complex topography with obstacles. Part II: Numerical predictions of fluid structures and benchmarking. Computers & Geosciences, Volume 73, 2014.
Round 2
Reviewer 1 Report
The authors have adequately addressed my comments. I recommend publication of the revised manuscript.
Reviewer 2 Report
The authors addressed all my previous queries. I thus recommend the publication of this manuscript in present form.